# SeLCA: Self-Supervised Learning of Canonical Axis

**Seungwook Kim\***[*]                                    WOOKIEKIM@POSTECH.AC.KR
**Yoonwoo Jeong\***                                    JEONGYW12382@POSTECH.AC.KR
**Chunghyun Park\***                                    P0125CH@POSTECH.AC.KR
**Jaesik Park**                                    JAESIK.PARK@POSTECH.AC.KR
**Minsu Cho**                                    MSCHO@POSTECH.AC.KR
*Pohang University of Science and Technology (POSTECH), South Korea*

**Editors:** Sophia Sanborn, Christian Shewmake, Simone Azeglio, Arianna Di Bernardo, Nina Miolane

## Abstract

Robustness to rotation is critical for point cloud understanding tasks as point cloud features can be affected dramatically with respect to prevalent rotation changes. In this work, we introduce a novel self-supervised learning framework, dubbed SeLCA, that predicts a canonical axis of point clouds in a probabilistic manner. In essence, we propose to learn rotational-equivariance by predicting the canonical axis of point clouds, and achieve rotational-invariance by aligning the point clouds using their predicted canonical axis. When integrated into a rotation-sensitive pipeline, SeLCA achieves competitive performances on the ModelNet40 classification task under unseen rotations. Our proposed method also shows high robustness to various real-world point cloud corruption presented by the ModelNet40-C dataset, compared to the state-of-the-art rotation-invariant method.
**Keywords:** SO(3)-equivariance, point cloud understanding, rotation invariance

## 1. Introduction

With recent advances in deep learning, there have been successful attempts to reason about 3D point clouds (Wang et al., 2019; Wu et al., 2019). However, consistent inference with respect to different rotations on point clouds remains challenging. Existing approaches to tackling this issue can be divided into two categories: a) learning to extract rotation-robust features using rotation-equivariant or invariant networks (Rao et al., 2019; Poulenard et al., 2019), and b) achieving rotation invariance by predicting a canonical pose via principal component analysis and aligning the point cloud to it (Kim et al., 2020; Li et al., 2021a).

In this paper, we focus on a major drawback of existing methods that has received little attention - their brittleness to real-world corruption, *e.g.*, occlusion and noise. Existing methods either have a strong reliance on the plane symmetries or the intrinsic geometries of point clouds (Li et al., 2021a; Xiao and Wachs, 2021), which are largely affected by point cloud corruption, therefore having limited applicability to real-world settings.

Our contribution can be summarized as follows:

- We introduce a novel self-supervised framework, dubbed SeLCA, that predicts a rotation-equivariant canonical axis of 3D point clouds.

- We propose a novel alignment scheme on geodesic spherical tessellations to effectively represent the canonical orientation distribution of a point cloud.

---

[*] \* denotes equal contribution

- We demonstrate the superior robustness of SeLCA against realistic point cloud corruption of the ModelNet40-C dataset, while being competitive on the ModelNet40 dataset for point cloud classification.

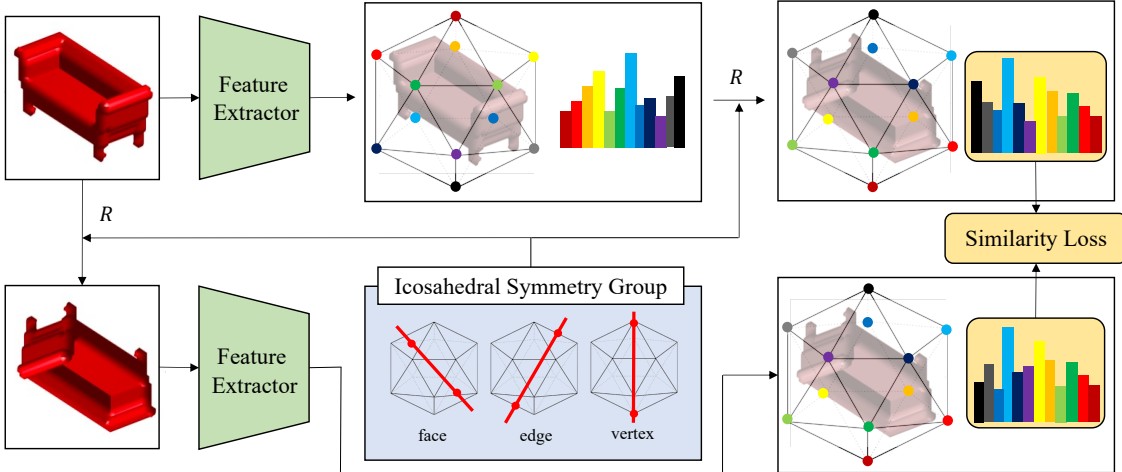

Figure 1: **Overview of the proposed SeLCA pipeline.** Our self-supervised pipeline trains the network by matching the histograms from a pair of rotated point clouds.

## 2. Self-Supervised Learning of Canonical Axis

It is impossible to provide strong supervision of canonical axes due to the absence of standard rules to pre-define the canonical axis of a point cloud. To this end, we train our SeLCA pipeline to output a canonical axis distribution that satisfies the following property: when a rotation $R$ is applied to the point cloud then the predicted canonical axis distribution should reflect the corresponding rotation by $R$, *i.e.* the canonical axis distribution should be equivariant to the rotation of point clouds. Figure 1 illustrates the overall training pipeline.

### 2.1. Geodesic Spherical Tessellation: Regular Icosahedron

We take inspiration from the axis-angle representation of 3D rotation to represent our canonical axis. As we want the output of our network to represent the canonical axis distribution, we need a discrete representation which can represent as many equispaced points on a unit sphere i.e., a geodesic spherical tessellation. We therefore use a regular icosahedron to represent our canonical axis distribution, noting the icosahedral group is the largest finite subgroup of SO(3).

### 2.2. Self-Supervised Training

First, we randomly select a rotation matrix from the icosahedral group to transform a point cloud. We feed the original point cloud and the rotated point cloud into our DGCNN feature extraction network to extract a 12-dimensional feature for each point cloud. We apply softmax on this feature to yield a 12-bin histogram, where each bin represents a vertex

Table 1: **Classification accuracy (%) on the ModelNet40 dataset.** The Acc. drop column represents the drop in accuracy from SO(3)/SO(3) to $z$/SO(3).

| | Method | Inputs | $z/z$ | $z$/SO(3) | SO(3)/SO(3) | Acc. drop |
|---|---|---|---|---|---|---|
| Rotation-sensitive | PointNet (Qi et al., 2017a) | xyz | 88.5 | 16.4 | 70.5 | 54.1 |
| | DGCNN (Wang et al., 2019) | xyz | 92.2 | 20.6 | 81.1 | 60.5 |
| | PointNet++ (Qi et al., 2017b) | xyz | 89.3 | 28.6 | 85.0 | 56.4 |
| | PointConv (Wu et al., 2019) | xyz | 91.6 | - | 85.6 | - |
| Rotation-equivariant | $a^3$SCNN (Liu et al., 2019) | voxel | 89.6 | 87.9 | 88.7 | 0.8 |
| | SFCNN (Rao et al., 2019) | xyz | 91.4 | 84.8 | 90.1 | 5.3 |
| | SFCNN (Rao et al., 2019) | xyz + normal | 92.3 | 85.3 | 91.0 | 5.7 |
| | RotPredictor (Fang et al., 2020) | xyz | 92.1 | - | 90.8 | - |
| Rotation-invariant | Li et al. (2021b) | feature | 89.4 | 89.4 | 89.3 | 0.1 |
| | RI-GCN (Kim et al., 2020) | xyz | 89.5 | 89.5 | 89.5 | 0.0 |
| | RI-GCN (Kim et al., 2020) | xyz + normal | 91.0 | 91.0 | 91.0 | 0.0 |
| | LGR-Net (Zhao et al., 2019) | xyz + normal + feature | 90.9 | 90.9 | 91.1 | 0.2 |
| | Li et al. (2021a) | xyz | 90.2 | 90.2 | 90.2 | 0.0 |
| | Ours ($k = 1$) | xyz | 88.4 | 86.1 | 87.0 | 0.9 |
| | Ours ($k = 3$) | xyz | 89.4 | 86.9 | 87.9 | 1.0 |

on a regular icosahedron, and each bin value represents the probability of a vector from the origin to the vertex to be the canonical axis of the point cloud. Note that we can map the selected rotation matrix (from the first step) to a permutation order of the histogram due to the closure of the icosahedral group. Then, we permute the histogram from the original point cloud using this permutation order, then supervise the two histograms - the permuted histogram from the original point cloud, and the histogram from the rotated histogram - to be close to one another, such that the canonical axis distribution of a point cloud is equivariant with respect to rotation. Formally, our loss is defined as:

$$\mathcal{L}(p_o, p_R) = -\log(\sum_{i=1}^{12} p_{o,i} p_{R,i}), \tag{1}$$

where $p_o$ and $p_R$ denote the output histogram bin of the original and rotated point clouds.

## 3. Experiments

### 3.1. Object Classification

We evaluate our model on the ModelNet40 dataset (Wu et al., 2015) on the task of classification. The results are presented in Table 1. We can see that our method achieves competitive performances compared to most rotation-equivariant methods and rotation-invariant methods despite not using equivariance-guaranteed architectures or representations. Also, we can see that the accuracy drop of our method from SO(3)/SO(3) to $z$/SO(3) is smaller than the majority of rotation-equivariant methods, showing that leveraging our learned rotational equivariance to model rotational invariance is empirically effective. We can also select up to top-$k$ canonical axis candidates to perform test-time augmentation *e.g.*, $k = 3$ denotes using a soft ensemble of output results from using top-$k$ argmax positions of the output canonical axis distribution. We guide the readers to the appendix for elaborate implementation details and evaluation settings.

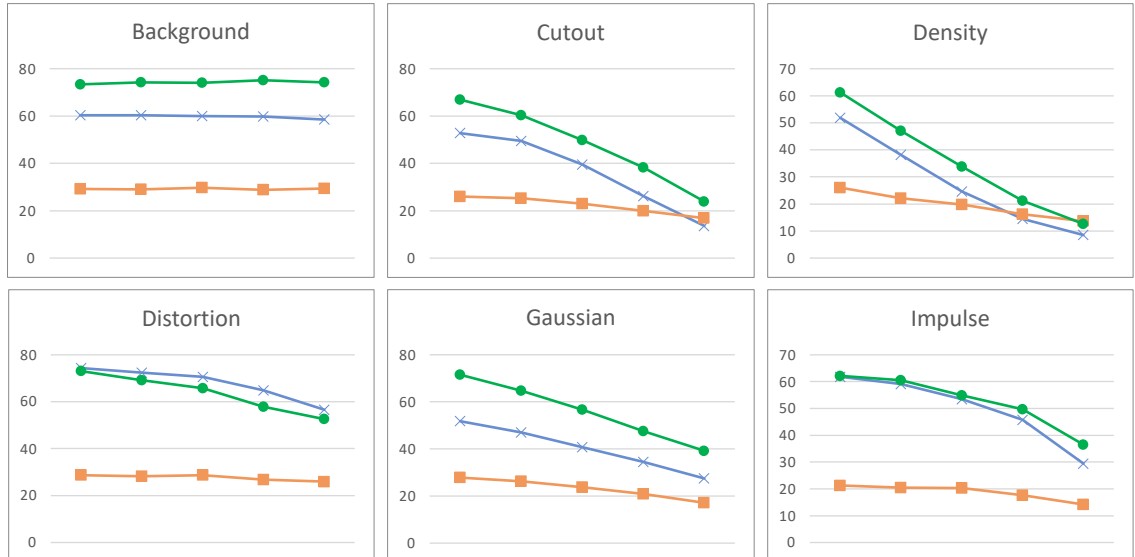

Figure 2: **Overall accuracy on the ModelNet40-C dataset.** We report the classification results of ours (green circle), Li et al. (2021a) (blue cross), and DGCNN (Wang et al., 2019) (orange square) with 6 different corruption and five severity levels increasing from left to right.

## 3.2. Classification on Noisy Real-world Datasets

We evaluate our method on the ModelNet40-C dataset (Sun et al., 2022) on the task of classification against DGCNN (Wang et al., 2019) and Li et al. (2021a), the current state-of-the-art PCA-based rotation-invariant method. The ModelNet40-C dataset is based on the ModelNet40 validation set, but augmented with 15 corruption types at 5 severity levels. We report the results for 6 corruption types in Figure 2. We refer the readers to the appendix for the full results. It can be seen that our method shows the highest performance overall under various types of point cloud corruption, where our robustness is especially pronounced under background, cutout and density corruption.

## 4. Discussion and Future Work

Albeit the rotational equivariance of our canonical axis distribution is not theoretically guaranteed as we do not rely on equivariant or invariant networks, the probabilistic nature of our approach enables our approach to be significantly more robust to various types of point cloud corruption compared to existing rotation-equivariant and -invariant representations, while yielding competent results on synthetic datasets. A promising future direction would be to build on our approach to improve the empirical rotational equivariance of the predicted canonical axis distribution. Another limitation of our approach is that we only predict the canonical axis of a point cloud, but not the canonical angle around the predicted canonical axis. Correctly predicting the canonical angle together with the canonical angle is expected to achieve a significant performance improvement, which we leave for future endeavors.

## Acknowledgments

This work was supported by Samsung Electronics Co., Ltd. and the Institute of Information & Communications Technology Planning & Evaluation (IITP) grants (No.2021-0-02068: AI Innovation Hub (50%), No.2022-0-00290: Visual Intelligence for Space-Time Understanding and Generation based on Multi-layered Visual Common Sense (40%), No.2019-0-01906: AI Graduate School Program at POSTECH (10%)) funded by Korea government (MSIT).

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

## Appendix A. Implementation Details

We employ DGCNN (Wang et al., 2019) as both our network structure to output the canonical pose distribution of a point cloud, and also as the base model for all downstream task experiments carried out in this paper. At train time, we first train our network in a self-supervised manner as explained in Section 2, and then train the downstream network while leaving our orientation estimator to be frozen. We add weight normalization (Salimans and Kingma, 2016) on the last layer during training. We use RAdam optimizer and cosine annealing learning rate scheduler with the initial learning rate 0.0005. The dropout ratio of DGCNN is set to 0.5 and the weight decay is set to 0.0001. We use centering (Caron et al., 2021) on each histogram bin. We randomly sample 1024 points from 2048 points of point clouds for each iteration. We follow the original training settings of DGCNN when training for downstream networks except for setting the initial learning rate to 0.01. For our experiments, we follow existing work (Li et al., 2021a) to compare different methods in 3 different ways: (1) $z/z$: both the train and test sets are rotated only around the $z$-axis, (2)$z/\mathrm{SO}(3)$: the train set is rotated only around the $z$-axis and the test set is randomly rotated, and (3) $\mathrm{SO}(3)/\mathrm{SO}(3)$: both the train and test sets are randomly rotated.

## Appendix B. Geometry-Aware Icosahedral Permutation

We go over the details of the geometry-aware icosahedral permutation, which is applied on the output of our ATP method in correspondence to the rotation applied on the point cloud, in order to provide self-supervision. The main idea here is that each histogram bin *i.e.*, entry in our normalized output, corresponds to a single vertex on the icosahedron. Therefore, applying a rotation sampled from the icosahedral rotational symmetry group to the icosahedron results in a permutation of the vertices *i.e.*, the histogram bins. This guarantees the rotated coordinates $\hat{v}$ are a permutation of $v$. For further details, refer to Algo 1.

---

**Algorithm 1** Geometry-aware Icosahedral Permutation

Original Histogram Bin: $\{h_i\}_{i=1}^{12}$
Coordinates of Icosahedron: $\{v_i\}_{i=1}^{12}$
Rotation Matrix $R$
**Output**: Permuted Histogram $\hat{h}$
**for** $i = \{1, 2, \cdots, 12\}$ **do**
    $\hat{v}_i = Rv_i$
    $j^* = \mathrm{argmax}_{j=1}^{12} ||\hat{v}_i - v_j||^2$
    $\hat{h}_i = h_{j^*}$
**end for**

---

## Appendix C. Full Results on the ModelNet40-C Dataset

In this section, we present the full set of results on the ModelNet40-C dataset, which accounts for a total of 3 settings ($z/z$, $z/\mathrm{SO}(3)$ and $\mathrm{SO}(3)/\mathrm{SO}(3)$), 15 corruption types and 5 severity levels.

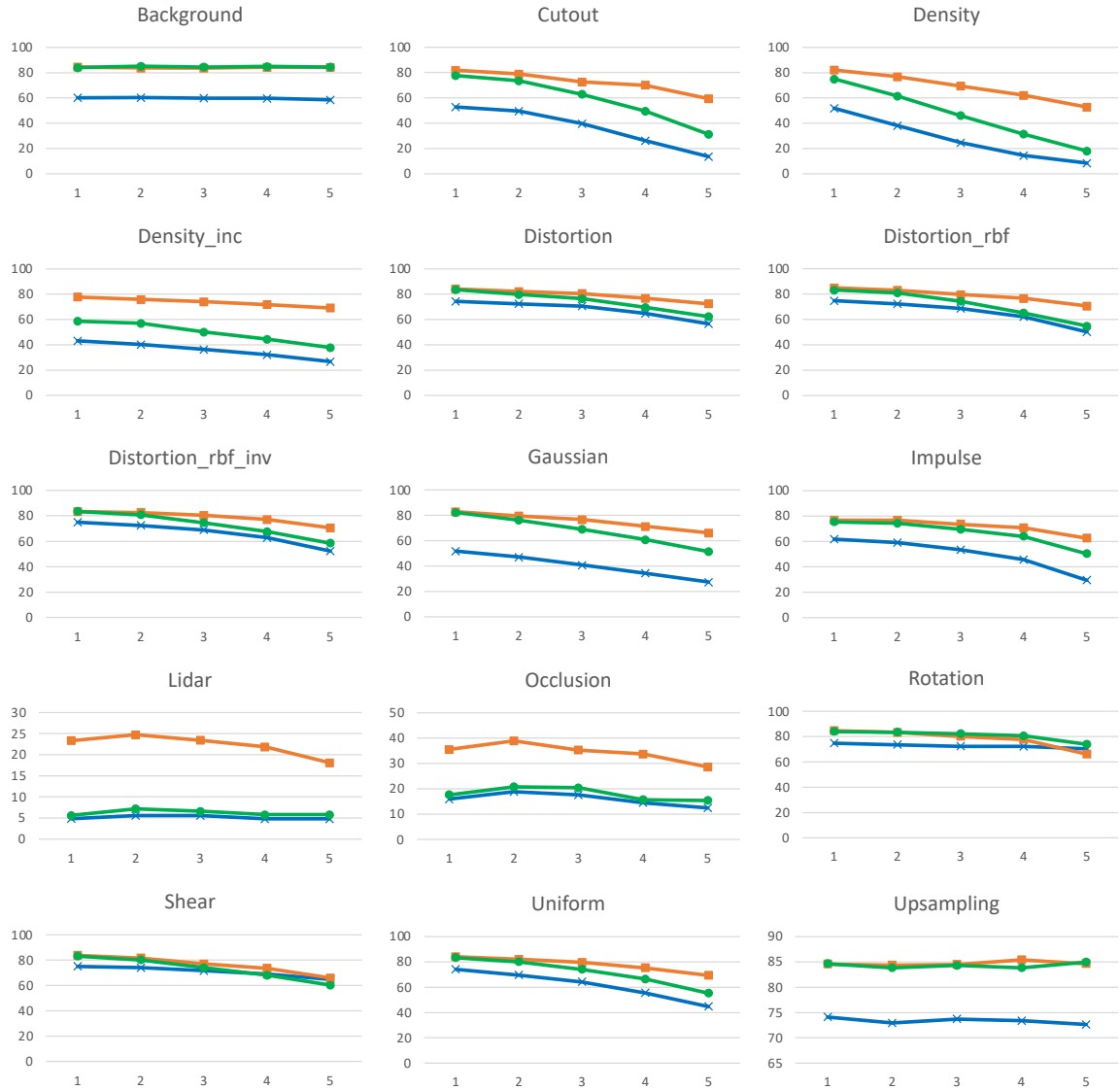

Figure 3: **Overall accuracy on the ModelNet40-C dataset using the ($z/z$) setting.** We report the classification results of ours (green circle), Li et al. (2021a) (blue cross), and DGCNN (Wang et al., 2019) (orange square) with 15 different corruption and five severity levels increasing from left to right. Under most corruption types, Li et al. (2021a) shows lower performances than other methods.

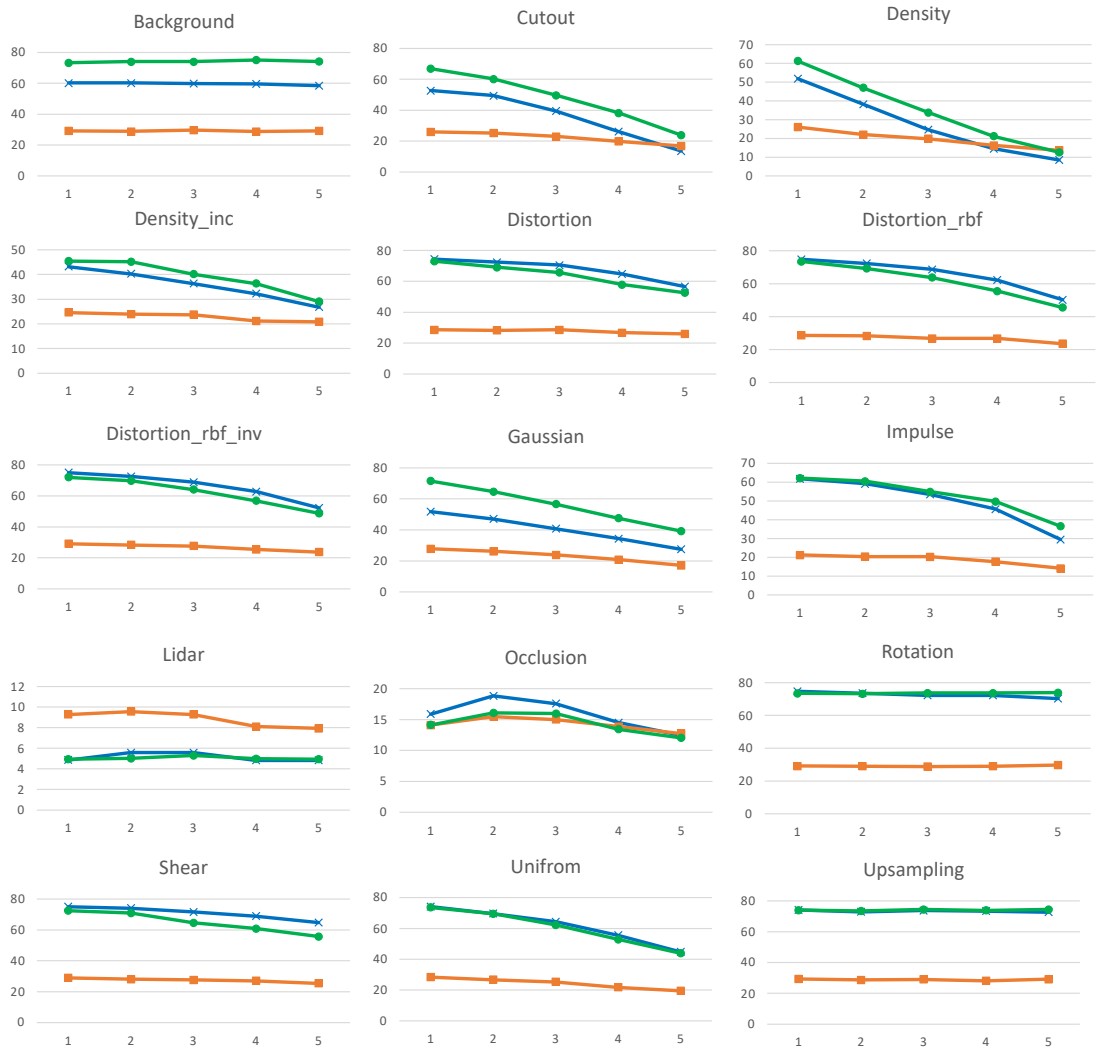

Figure 4: **Overall accuracy on the ModelNet40-C dataset using the ($z$/**SO**(3))
setting.** We report the classification results of ours (green circle), Li et al.
(2021a) (blue cross), and DGCNN (Wang et al., 2019) (orange square) with 15
different corruption types and five severity levels increasing from left to right. In
most corruption types, DGCNN shows much lower classification accuracy than
the others. Ours shows the best performance overall.

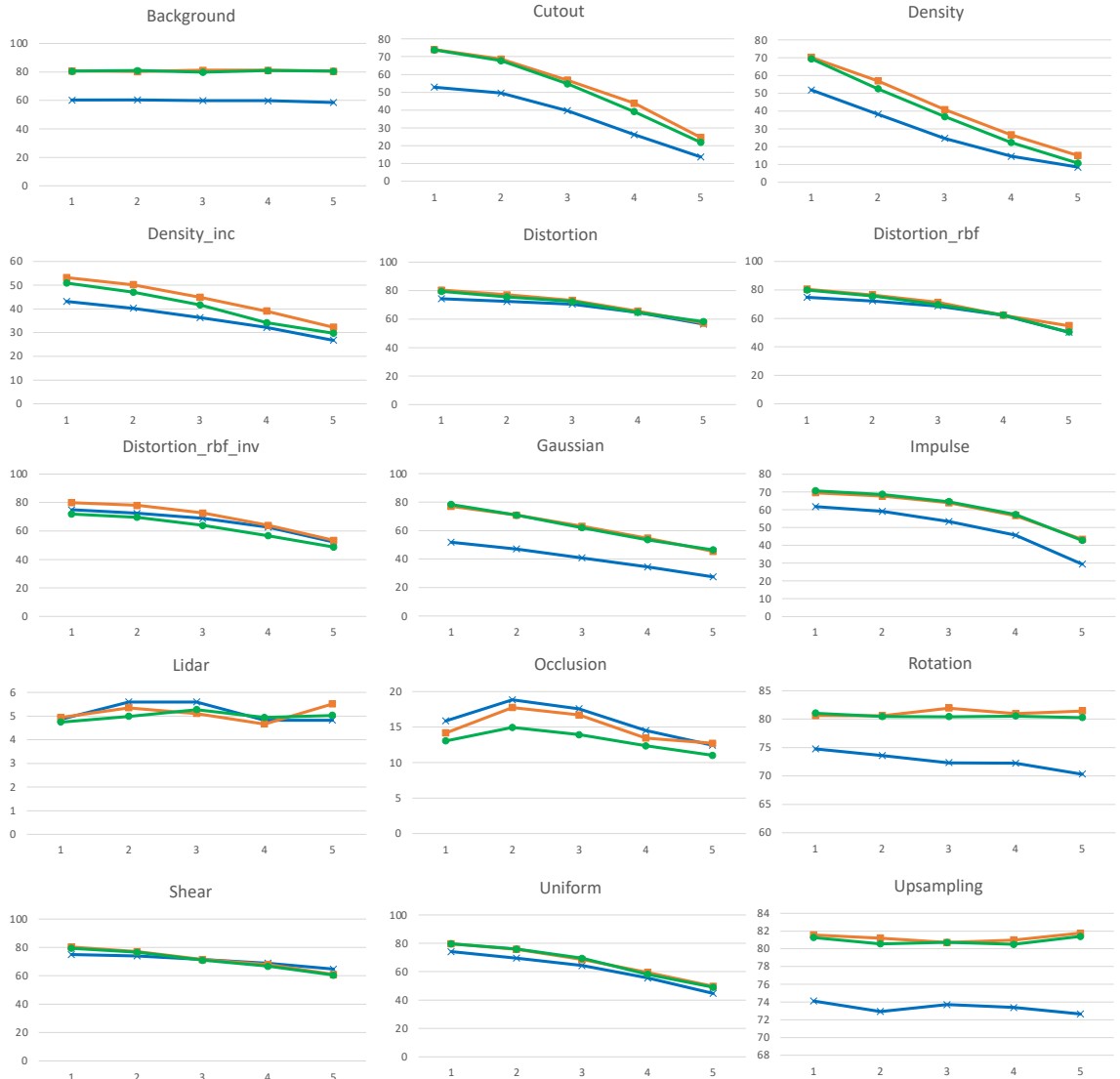

Figure 5: **Overall accuracy on the ModelNet40-C dataset using the (SO(3)/SO(3)) setup.** We report the classification results of ours (green circle), Li et al. (2021a) (blue cross), and DGCNN (Wang et al., 2019) (orange square) with 15 different corruption types and five severity levels increasing from left to right. DGCNN and ours show similar performances, while Li et al. (2021a) shows much lower classification performances under several corruption types.

