# OpenReview forum: "SeLCA: Self-Supervised Learning of Canonical Axis"
_NeurIPS.cc/2022/Workshop/NeurReps — NeurReps 2022 Poster_

### Official Review · Reviewer_imzP · 2022-10-14
**Mostly great, clean-up figures**

**Confidence:** 3
**Soundness:** 4
**Presentation:** 2
**Contribution:** 4
**Overall Rating:** 8

**Summary:**

Finding the canonical axis of point clouds is important for classification and object comprehension tasks since it allows instances to be transformed into a rotationally invariant representation if paired with a canonical angle. The authors have developed a method for learning the canonical axis through a self-supervised manner that is proposed to be more robust to occlusion, cutout, and background noise that is often present in real-world data. This is accomplished by extracting a 12-dimensional feature, using DGCNN, on the original point cloud and a rotation of the original point cloud. Soft-max is applied to both features to create two 12-bin histograms. The rotation matrix that was applied to the original point-cloud data can now be mapped to a permutation order and this permutation order can be applied to the histogram of the original point-cloud. A similarity loss function is then used to between these two histograms. The values of each bin represent the probability that that vector from the origin to the vertex on the icosahedral is the canonical axis. Therefore, the bin with the largest magnitude is the predicated canonical axis. This proposed method performs competitively with other current methods on clean data sets and performs especially well on noisy real-world datasets.

**Questions:**

How does the feature-extraction network affect results? Could others be used?

How fast is it trained compared to other methods? Memory usage? System requirements?


**Limitations:**

No mention of how fast their method is compared to others. This is important for real-world application which this method is supposed to be for. Authors do mention that predicting the canonical angle would mostly likely help performance and is out of the scope of this paper.

**Recommended Decision:**

3: Accept

**Relevance:**

3: Solid fit

**Strengths And Weaknesses:**

Originality: Not the most original but the use of a regular icosahedron from feature extraction to represent canonical probability distributions is clever.

Quality: Provides benchmark data of various other methods as well as the proposed method in ModelNet40 and ModelNet40-C. While Table 1 is well labelled, Figure 2 is poorly labelled. Technically sound.

Clarity: It is mostly well-organized and well written. The Icosahedral Symmetry Group operation block should be better placed to more clearly  show that it is applied to the original point cloud and is converted to a permutation order to be applied to the original point-cloud histogram. Right now the tees in the arrows create some confusion. Figure 2 should have axes labels and legends that don't rely on the captions.

Significance: Point-cloud representations are of high-significance in many fields including robotics, navigation, and pose estimation. Creating methods for real-world applications is of utmost importance.

**Submission Track:**

Extended Abstract (4 Page)

---

### Official Review · Reviewer_NUkW · 2022-10-18
**Great idea but can only be applied to clean data**

**Confidence:** 4
**Soundness:** 2
**Presentation:** 3
**Contribution:** 2
**Overall Rating:** 4

**Summary:**

The authors proposed a self-supervised pretext task of predicting the canonical axis of a point cloud. The goal is to provide rotation invariance and equivariance and reduce data sample complexity.

**Questions:**

- Can the authors explain why the proposed approach is robust to occlusion and noise?
- Does this approach work for other datasets that contains such noise, e.g. ScanObjectNN?

**Limitations:**

- According to my understanding, the work is limited to single-object point cloud that are relatively "clean", i.e. densely sampled, no occlusion, no significant sparsity.

**Recommended Decision:**

1: Reject

**Relevance:**

3: Solid fit

**Strengths And Weaknesses:**

Strengths:
- The task of designing rotation invariant and equivariant neural networks is of great importance. A number of recent works have been focusing on this topic.
- Designing self-supervised pretext task for point cloud is still under explored. The authors proposed a simple and effective pretext task.

Weakness:
- The robustness against occlusion and noise to the point cloud is unfounded to me. The paper didn't explain why the approach is robust to these disturbances.

**Submission Track:**

Extended Abstract (4 Page)

---

### Official Review · Reviewer_1puj · 2022-10-18
**Fair paper, some key details are missing**

**Confidence:** 4
**Soundness:** 2
**Presentation:** 3
**Contribution:** 2
**Overall Rating:** 5

**Summary:**

The paper introduces SeLCA, a self-supervised approach to learn the reference frame of a point cloud. The learned reference frame can then be used to align different point clouds before feeding them to other methods to achieve rotation invariance. The validity of the proposed method is tested with the shape classification task on the ModelNet40 and ModelNet40-C datasets. On ModelNet40 SeLCA underperforms competitor approaches, on ModelNet40-C it outperforms the state-of-the-art PCA-based rotation-invariant method proposed by Li et al. 2021 and DGCNN in the z/SO(3) setting.

**Questions:**

Too many information is given for granted. Here follows some examples.
- When performing classification are you applying first the pre-trained self-supervised network to estimate its canonical axis, rotate the input shape accordingly, and then apply an exiting deep classifier? If this is the process, then I ask the authors to specify which deep classifier they used in Table 1. Potentially it can be shown how all the approaches listed in the rotation-sensitive category behaves after the canonical axis alignment learned by SeLCA. Otherwise I ask the authors to better specify their setting.
- It's not clear what the experimental setting is. What do you mean by z/z, z/SO(3), and SO(3)/SO(3)? My understanding is that it refers to the rotations applied at train and test time, where z denotes no rotation and SO(3) a random rotation. I couldn't find any information regarding the experimental setup, making it difficult to judge the quality of the results reported.
- Please add ticks to the x axis in Figure 2 to better interpret the results knowing the intensity of the perturbation.


**Limitations:**

As limitation the authors identify the prediction of the canonical axis only, without predicting the canonical angle associated.

**Recommended Decision:**

2: Borderline

**Relevance:**

3: Solid fit

**Strengths And Weaknesses:**

Strenghts: The method developed to learn the canonical axis of object point clouds is very interesting.

Weaknesses: Important details are missing to properly evaluate its effectiveness. Experiments do not show a clear improvement over competitors. On the ModelNet40 dataset SeLCA underperforms rotation invariant approaches and behaves similarly to rotation equivariant methods. Not enough details are provided to appreciate why SeLCA should still be appealing for the community. On ModelNet40-C the comparison is limited to the state-of-the-art PCA-based rotation-invariant method proposed by Li et al. 2021 and DGCNN. SeLCA outperforms both competitors on the z/SO(3) setting (the selection of results presented in the main paper) but underperforms DGCNN in the z/z setting and performs on pair with the two competitors in the SO(3)/SO(3) settings presented in the appendix (z/z ).

**Submission Track:**

Extended Abstract (4 Page)

---

### Decision · Program_Chairs · 2022-10-21

Accept (Poster)